# DOMAIN-AGNOSTIC FEW-SHOT CLASSIFICATION BY LEARNING DISPARATE MODULATORS

## ABSTRACT

Although few-shot learning research has advanced rapidly with the help of meta-learning, its practical usefulness is still limited because most of the researches assumed that all meta-training and meta-testing examples came from a single domain. We propose a simple but effective way for few-shot classification in which a task distribution spans multiple domains including previously unseen ones during meta-training.

The key idea is to build a pool of models, each of which is responsible for a different group of tasks, and to learn to select the best one for a particular task through multi-domain meta-learning. This reduces task-specific adaptation over a complex task distribution to a simple selection problem rather than modifying the model with a number of parameters at meta-testing time. Inspired by common multi-task learning techniques, we let all models in the pool share a base network and add a separate modulator to each model to refine the base network in its own way. This architecture allows the pool to maintain representational diversity and each model to have domain-invariant representation as well.

Experiments show that our selection scheme outperforms other few-shot classification algorithms when target tasks come from many different domains. They also reveal that aggregating outputs from all constituent models is effective for tasks from unseen domains indicating the effectiveness of our framework.

## 1 INTRODUCTION

Few-shot learning in the perspective of meta-learning aims to train models which can quickly solve novel tasks or adapt to new environments with limited number of examples. In case of few-shot classification, models are usually evaluated on a held-out dataset which does not have any common class with the training dataset. In the real world, however, we often face harder problems in which novel tasks arise arbitrarily from many different domains even including previously unseen ones.

In this study, we propose a more practical few-shot classification algorithm to generalize across domains beyond the common assumption, i.e., meta-training and meta-testing within a single domain. Our approach to cover a complex multi-domain task distribution is to construct a pool of multiple models and learn to select the best one given a novel task through meta-training over various domains. This recasts task-specific adaption across domains as a simple selection problem, which could be much easier than manipulating high-dimensional parameters or representations of a single model to adapt to a novel task.

Furthermore, we enforce all models to share some of the parameters and train per-model modulators with model-specific parameters on top of that. By doing so, each model could keep important domain-invariant features while the model pool has representational diversity as a whole without a significant increase of model parameters.

We train and test our algorithms on various image classification datasets with different characteristics. Experimental results show that the proposed selection scheme outperforms other state-of-the-art algorithms in few-shot classification tasks from many different domains without being given any knowledge of the domain which the task belongs to. We also show that even few-shot classification tasks from previously unseen domains, i.e., domains which have never appeared during meta-training, can be done successfully by averaging outputs of all models.

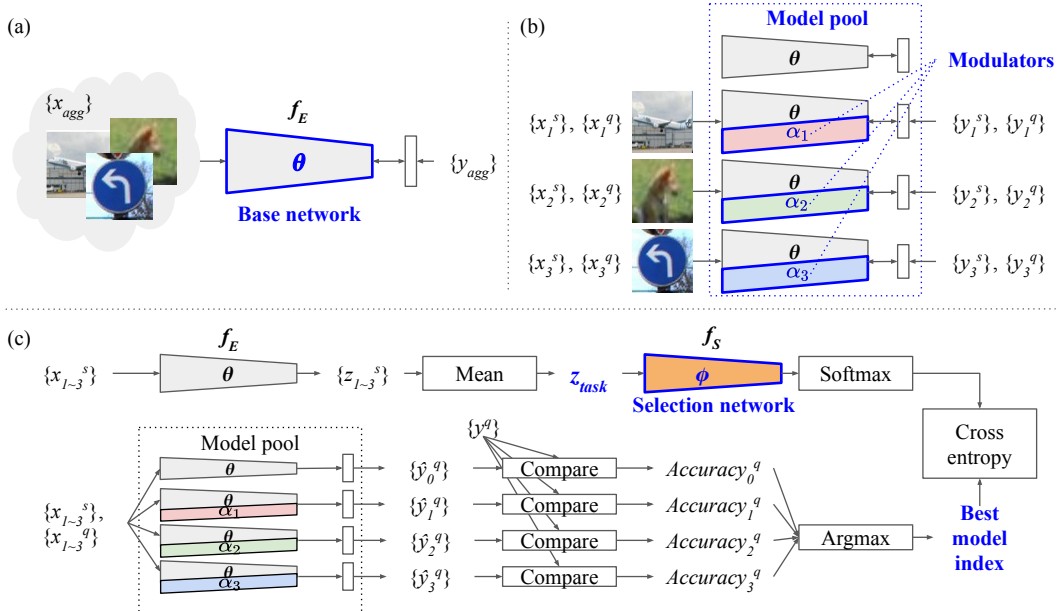

Figure 1: Training (a) a base network $f_E(\cdot; \theta, \cdot)$, (b) modulators $f_E(\cdot; \cdot, \alpha_i)$, (c) a selection network $f_S(\cdot; \phi)$. ($x_m^n$: an example, $y_m^n$: a label, $\hat{y}_m^n$: a prediction, $z_m^n$: an embedded vector of a support ($n=s$) or query ($n=q$) set in a domain $m$.)

## 2 METHODS

### 2.1 PROBLEM STATEMENT

We follow the common setting of few-shot classification in the meta-learning community (Vinyals et al., 2016). For a $N$-way, $K$-shot classification task, an episode which consists of a support set $S = \{(x_i^s, y_i^s)\}_{i=1}^{NK}$ and a query set $Q = \{(x_i^q, y_i^q)\}_{i=1}^{T}$ is sampled from a given dataset, where $x_i^s$, $x_i^q$, $y_i^s$ and $y_i^q$ represent examples and their correct labels respectively and $T$ is the number of query examples. Once a model has been trained with respect to a number of random episodes at meta-training time, it is expected to predict a correct label for an unlabeled query given only a few labeled examples (i.e., support set) even if all these came from classes which have never appeared during meta-training.

Based on this setting, we try to build a domain-agnostic meta-learner beyond the common meta-learning assumptions, i.e., meta-training within a single domain and meta-testing within the same domain. We perform meta-training over multiple diverse domains, which we call source domains $D_{S_1}, D_{S_2}, \cdots, D_{S_M}$, where $M$ is the number of source domains, expecting to obtain a domain-generalized meta-learner. Since we presume that one particular dataset defines its own domain, we realize this idea by training this meta-learner on various tasks from many different datasets.

In our study, the trained meta-learner is meta-tested on a target domain $D_T$ for two types of cross-domain few-shot classification tasks. One is a task which is required to classify from held-out classes of multiple source domains (i.e., $D_T \in \{D_{S_1}, D_{S_2}, \cdots, D_{S_M}\}$) without knowing from which dataset each task is sampled. This could be used to evaluate whether the meta-learner is capable of adapting to a complex task distribution across multiple domains. We also tackle a task sampled from previously unseen datasets during the meta-training (i.e., $D_{S_i} \cap D_T = \emptyset$ for all $i$), which requires to generalize over out-of-distribution tasks in domain-level.

### 2.2 BUILDING A POOL OF EMBEDDING MODELS WITH DISPARATE MODULATORS

Basically, we perform metric-based meta-learning to learn a good metric space in which the support and query examples from the same class are located closely over various domains. While recent meta-learning methods have been proposed to train a single model commonly applicable to various

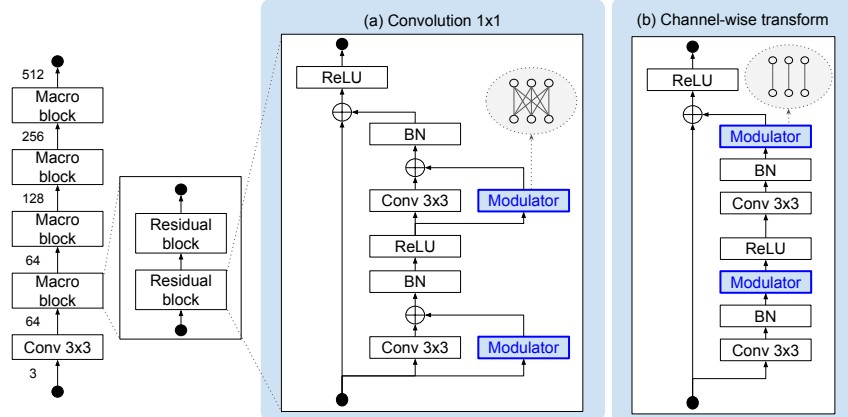

Figure 2: Architecture of one embedding model $f_E(\cdot; \theta, \alpha_i)$.

potential tasks and to learn to adjust the model to a particular task for further improvement (Rusu et al., 2019; Oreshkin et al., 2018; Ye et al., 2018; Triantafillou et al., 2019), we train a pool of multiple embedding models each of which defines a different metric space and a meta-model to select the best model from them given a novel task. This makes task-specific adaptation easier and more effective to learn because our approach is required to solve only a simple classification problem, i.e., choose one of all pre-trained models, instead of learning to manipulate high-dimensional model components, such as model parameters or activations, directly to adapt to novel tasks from various domains.

Rather than training each model separately, we take an approach that all models share a considerable amount of parameters and they are differentiated by adding per-model modulators as done usually in multi-task learning (Ruder, 2017). The rationale behind this is to let our model pool capture good domain-invariant features by the shared parameters as well as have diversity, which is desirable to represent the complex cross-domain task distribution, without a significant increase of the number of parameters.

To realize this idea, we first build a base network $f_E(\cdot; \theta)$ shared among all models. One large virtual dataset is constructed by aggregating all source datasets. The base network is trained on this dataset following typical supervised learning procedure (Figure 1(a)). In the next step, we build one model per source domain by adding a per-model modulator with a parameter set $\alpha_i$ on top of the frozen base network. We then train each modulator on one dataset $D_{S_i}$ by performing metric-based meta-learning in the same way as the Prototypical Networks (*ProtoNet*) (Snell et al., 2017) (Figure 1(b)). Finally, we have a pool of embedding models which are ready for non-parametric classification in the same way as *ProtoNet*.

As shown in Figure 2, we add the modulator components to the base network in a per-layer basis following the idea proposed in (Rebuffi et al., 2018). This way has turned out to be more effective than the conventional approach, i.e., adding a few extra layers for each model, for domain-specific representation. Moreover, this allows each modulated model to have the same computational cost at inference time as the base network's because all modulator components can be fused into existing convolution $3\times3$ operations. We try two modulator architectures, convolution $1\times1$ and channel-wise transform (i.e., per-channel trainable scale and bias). The former shows slightly better performance whereas the latter uses much less parameters only incurring negligible memory overhead to the pool. More details of the architecture including the number of parameters can be found in Appendix B.

## 2.3 LEARNING TO SELECT THE BEST MODEL FOR A TARGET TASK

After the construction of the pool, we build a meta-model to predict the most suitable model from all constituent models in the pool for a given task as the final step of our training. By training this model over a number of episodes sampled randomly from all available source datasets, we expect this ability to be generalized to novel tasks from various domains including unseen ones during meta-training.

As depicted in Figure 1(c), this meta-model parameterized by $\phi$, which we call a model selection network, is trained in order to map a task representation $z_{task}$ for a particular task to an index of the best model in the model pool. The task representation is obtained by passing all examples in the support set of the task through the base network and taking a mean of all resulting embedded vectors to form a fixed-length summary of the given task. During meta-training, the index of the best model, which is the ground truth label for training the selection network, is generated by measuring the classification accuracy values of all models in the pool given the query set and picking an index of one which has the highest accuracy.

In our setup, the task-specific adaptation is reduced to a $(M+1)$-way classification problem when we have $M+1$ embedding models including the base network learned from $M$ available source datasets. Learning this classifier could be far simpler than learning to adapt model parameters (Rusu et al., 2019), embedded vectors (Ye et al., 2018) or per-layer activations (Oreshkin et al., 2018) to a particular task because their dimensions are usually much larger than that of our selection network outputs, i.e., the number of the pre-trained models.

The overall training procedure for constructing the pool and training the selection network is summarized in Algorithm 1 in Appendix C.

## 2.4 INFERENCE WITH THE POOL AT META-TESTING TIME

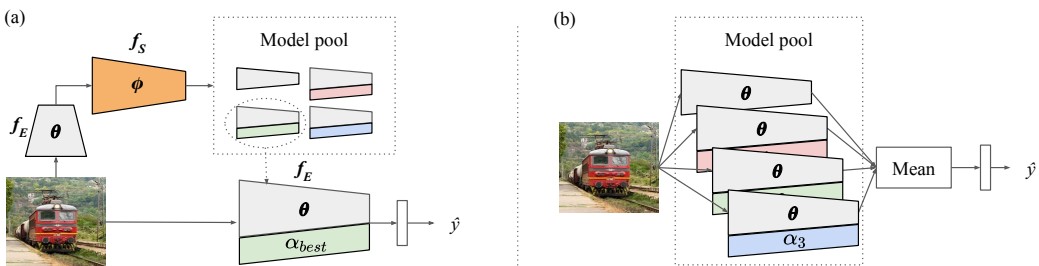

Figure 3: Inference (a) with one model chosen by the selection network $f_S(\cdot; \phi)$ (b) by averaging output probabilities from all constituent models in the pool.

One way of the inference is to predict a class with the best single model chosen by the selection network $f_S(\cdot; \phi)$ for a given support set (Figure 3(a)). Following the method proposed in *ProtoNet* as shown in Equation 1, a class prediction $\hat{y}$ to a query example $x^q$ is made by finding a class whose prototype $c_y$ is the closest one to an embedded vector of the given query example. Specifically, the prototype for the class $y$ is defined as a mean of embedded vectors of all examples in the support set belonging to a class $y$ and squared Euclidean distances $d_i^y$ to these prototypes are compared between the classes.

$$\hat{y} = \operatorname*{argmin}_{y} d_i^y(x^q) = \operatorname*{argmin}_{y} \|c_y - f_E(x^q; \theta, \alpha_i)\|^2. \tag{1}$$

Another way to benefit from the model pool is to combine outputs from all available embedding models for inference (Figure 3(b)). We take the simplest approach to average outputs of all models in the level of class prediction probability. As described in Equation 2, we collect output probabilities $p(y \mid x^q; i)$ based on the relative distances to class prototypes $d_i^y$ for a given task. Then, our final prediction $\hat{y}$ would be a class to maximize a mean of these probabilities from all $M + 1$ models.

$$\hat{y} = \operatorname*{argmax}_{y} \frac{1}{M+1} \sum_{i=0}^{M} p(y \mid x^q; i) = \operatorname*{argmax}_{y} \frac{1}{M+1} \sum_{i=0}^{M} \operatorname{softmax}(-d_i^y(x^q)) \tag{2}$$

As the last step, we adopt test-time 'further adaptation' proposed in (Chen et al., 2019), which turned out to make additional performance improvement in most cases. Both experimental results with and without the further adaptation are presented in Section 3 and Appendix D with its implementation details in Appendix C.

# 3 EXPERIMENTS

## 3.1 SETUP

### 3.1.1 DATASETS

We use eight image classification datasets, denoted as Aircraft, CIFAR100, DTD, GTSRB, ImageNet12, Omniglot, SVHN, UCF101 and Flowers, introduced in the *Visual Decathlon Challenge* (Rebuffi et al., 2017) for evaluation. These are considered as eight different domains in our experiments. All datasets are resplit for the few-shot classification, i.e., no overlap of classes between meta-training and meta-testing. More details about the datatsets can be found in Appendix A.

### 3.1.2 ALGORITHMS

We denote our methods using the model picked by the selection network as *DoS* (Domain-generalized method by Selection) and *DoS-Ch*, which modulate the base network with convolution $1\times1$ and channel-wise transform respectively. We also explore our averaging-based methods, *DoA* (Domain-generalized method by Averaging) and *DoA-Ch*, which generate an output by averaging class prediction probabilities of all constituent models modulated by the proposed two types of modulators.

Our algorithms are compared with *Fine-tune*, *Simple-Avg*, *ProtoNet* (Snell et al., 2017), *FEAT* (Ye et al., 2018) and *ProtoMAML* (Triantafillou et al., 2019). *Fine-tune* is a baseline method, which adds a linear classifier on top of the pre-trained base network and fine-tune it with the support set examples for 100 epochs at meta-testing. In *Simple-Avg*, we train an embedding model independently on each source domain without sharing any parameters and perform inference by averaging class prediction probabilities of all these models. *FEAT* and *ProtoMAML* are the state-of-the-art algorithms focusing on single domain and cross-domain setups respectively. *TADAM* (Oreshkin et al., 2018) was also tested but excluded from the results because its training did not converge in our setup. All these algorithms are tested by our own implementations.

### 3.1.3 TRAINING AND TESTING

We pre-train all models of the compared algorithms with our base network since the pre-training has shown additional performance gain empirically. Then, we meta-train these models in an algorithm-specific way on randomly generated episodes except the *Fine-tune*, the non-episodic baseline. At each episode, a target domain is chosen randomly from source domains, then a target task is sampled from that domain. The trained models are tested for 5-way classification tasks with 1-shot and 5-shot configurations.The test results are averaged over 600 test episodes with 10 queries per class. Other details about the training are explained in Appendix C.

## 3.2 RESULTS

### 3.2.1 FEW-SHOT CLASSIFICATION ON SEEN DOMAINS

We evaluate the above-mentioned algorithms in a multi-domain test setup constructed using all available datasets. Specifically, we meta-train a model for each algorithm on all available eight datasets. Then, we meta-test the trained model for various tasks sampled from these eight datasets without knowing which dataset the task comes from. Figure 4 depicts test accuracy for each target dataset and the average accuracy over the eight datasets.

Our selection methods, *DoS* and *DoS-Ch*, outperform other few-shot classification methods in most cases. Two state-of-the art algorithms, *FEAT* and *ProtoMAML*, do not seem as effective as ours under this complex task distribution across domains. *ProtoMAML* shows comparable or better results in some cases, but much inferior results in other cases. *ProtoNet* seems relatively stable, but does not produce better results in any case. *FEAT* works worse than these two algorithms in most cases.

Although our averaging methods, *DoA* and *DoA-Ch*, seem competitive in many cases, they are out-performed by the selection methods always. This implies that the learned selection network is working properly, which is highly likely to select the model with the modulator trained on the same domain as the given task even without any information about the domain at testing time. Another

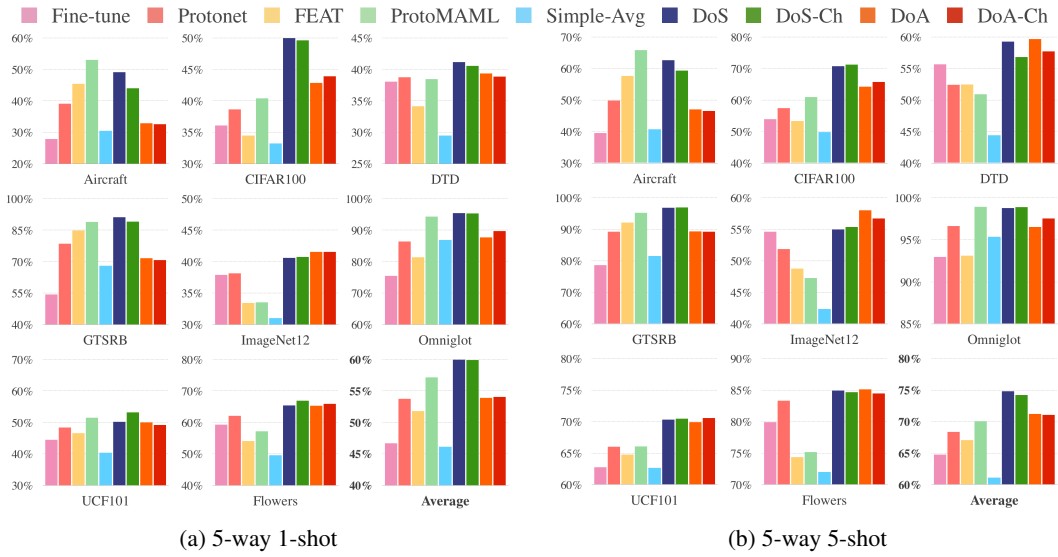

(a) 5-way 1-shot

(b) 5-way 5-shot

Figure 4: Test accuracy on various seen domains. (Accuracy values are shown in Table A5.)

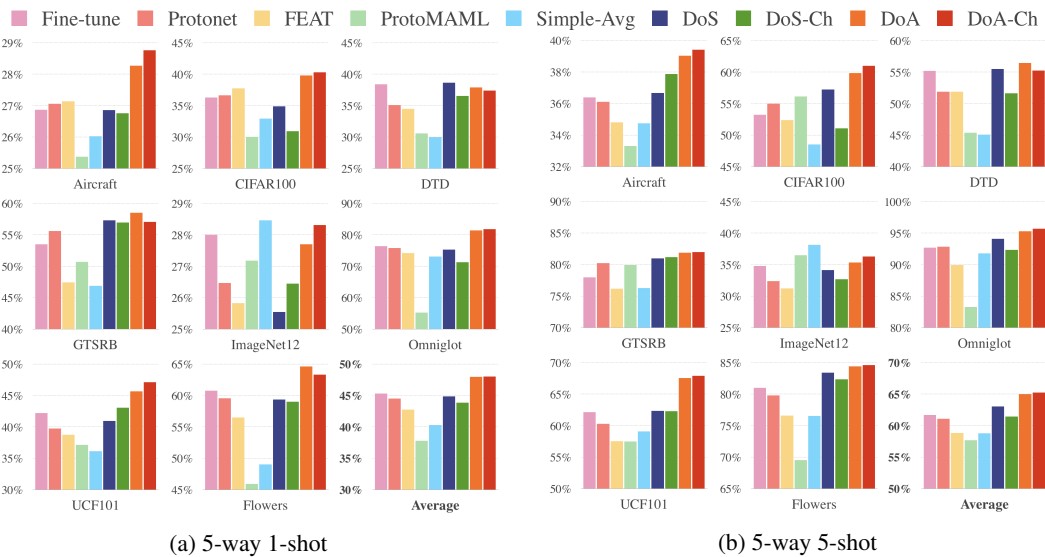

(a) 5-way 1-shot

(b) 5-way 5-shot

Figure 5: Test accuracy on various unseen domains. (Accuracy values are shown in Table A6.)

implication is that the best single model might be better than the averaging approach if a model from the same domain exists. It is also worth noting that *DoS-Ch* is quite competitive despite much less number of parameters than *DoS*.

### 3.2.2 FEW-SHOT CLASSIFICATION ON UNSEEN DOMAINS

We also report the results on unseen domains in Figure 5. Given a target dataset for test, we train all models on other seven datasets. Therefore, we end up with eight different models for each algorithm because there exist eight different combinations of the source datasets.

Our methods still outperform other algorithms in this more challenging setting. This reveals that our approach can be generalized to novel domains as well. Differently from the seen domain cases, our averaging methods, *DoA* and *DoA-Ch*, perform better than all other algorithms including our selection methods. It seems to make sense since the averaging could induce natural synergy between beneficial models even if we do not know which models are good for a given task. However,

our averaging methods significantly outperform *Simple-Avg*, which implies that our way of the pool construction to encourage keeping domain-invariant features is another key factor to the high performance of our averaging methods.

### 3.2.3 FEW-SHOT CLASSIFICATION ON VARYING NUMBER OF SOURCE DOMAINS

We conduct experiments with varying number of source datasets. Following the common real-world situation, we add from the largest dataset to the smallest one to our sources for meta-training. Tables 1 and 2 show the experimental results with 2, 4, and 6 source datasets on seen and unseen domains respectively.

Our selection and averaging methods outperform others consistently on seen and unseen domains similarly to the previous results. Apart from comparing between the algorithms, it is commonly observed over all algorithms that the added source often harms the performance. For example, the CIFAR100 tasks tend to work poorer as the number of source datasets increases in the seen domain case. This means that we should pay more attention to avoiding negative transfer between heterogeneous domains.

Table 1: Few-shot classification accuracy of varying number of sources on seen target domains.

| S | T | METHODS | | | | | | | |
|---|---|---|---|---|---|---|---|---|---|
| | | FINE-TUNE | PROTONET | FEAT | PROTOMAML | DOS | DOS-CH | DOA | DOA-CH |
| C,I | C | 54.72% | 65.47% | 65.39% | 69.51% | **73.04**% | 72.63% | 71.19% | 69.58% |
| | I | 57.50% | 55.88% | 52.53% | 56.37% | 57.71% | 57.84% | **59.39**% | 58.05% |
| AVERAGE | | 56.11% | 60.68% | 58.96% | 62.94% | **65.38**% | 65.24% | 65.29% | 63.82% |
| C,G,I,O | C | 54.24% | 58.31% | 63.82% | 68.93% | **71.11**% | 70.15% | 68.14% | 66.67% |
| | G | 81.21% | 90.58% | 95.40% | 96.61% | **96.96**% | 96.36% | 92.94% | 91.55% |
| | I | 55.70% | 52.04% | 50.58% | 53.72% | 55.78% | 56.10% | **58.04**% | 57.74% |
| | O | 94.82% | 97.14% | 95.10% | **99.52**% | 98.94% | 98.73% | 97.86% | 97.64% |
| AVERAGE | | 71.49% | 74.52% | 76.23% | 79.70% | **80.70**% | 80.33% | 79.25% | 78.40% |
| A,C,G, I,O,U | A | 40.68% | 51.92% | 62.77% | **70.78**% | 61.94% | 57.49% | 47.77% | 46.86% |
| | C | 54.03% | 58.68% | 56.56% | 60.47% | 69.54% | **70.32**% | 66.48% | 65.41% |
| | G | 76.35% | 89.95% | 95.64% | 96.50% | **97.71**% | 96.78% | 89.65% | 89.62% |
| | I | 53.37% | 52.80% | 48.62% | 51.74% | 54.59% | 56.03% | 57.34% | **58.03**% |
| | O | 94.06% | 96.94% | 95.30% | **99.11**% | 98.89% | 98.76% | 97.20% | 97.92% |
| | U | 62.01% | 65.49% | 62.20% | 67.11% | 70.44% | 70.23% | 69.16% | **71.42**% |
| AVERAGE | | 63.42% | 69.30% | 70.18% | 74.29% | **75.52**% | 74.94% | 71.27% | 71.54% |

S: source datasets, T: target dataset
A: Aircraft, C: CIFAR100, G: GTSRB, I: ImageNet12, O: Omniglot, U: UCF101, F: Flowers

Table 2: Few-shot classification accuracy of varying number of sources on unseen target domains.

| S | T | METHODS | | | | | | | |
|---|---|---|---|---|---|---|---|---|---|
| | | FINE-TUNE | PROTONET | FEAT | PROTOMAML | DOS | DOS-CH | DOA | DOA-CH |
| C,I | A | 38.65% | 39.38% | 36.71% | 35.37% | 38.43% | 39.27% | 39.14% | **40.57**% |
| | D | 53.89% | 55.27% | 52.39% | 51.04% | 54.80% | 54.90% | **57.15**% | 56.99% |
| | G | 70.01% | 81.37% | 77.67% | 81.25% | **83.83**% | 82.65% | 80.13% | 79.93% |
| | O | 92.72% | 92.62% | 90.61% | 91.37% | 93.11% | 93.31% | 94.00% | **94.34**% |
| | U | 63.80% | 63.25% | 60.04% | 59.89% | 63.67% | 64.81% | 66.28% | **67.15**% |
| | F | 79.28% | 81.77% | 78.86% | 80.50% | 81.26% | 80.84% | **82.96**% | 82.85% |
| AVERAGE | | 66.39% | 68.94% | 66.05% | 66.63% | 69.18% | 69.30% | 69.94% | **70.31**% |
| C,G,I,O | A | 38.12% | 36.88% | 35.11% | 34.96% | 37.90% | 38.04% | 39.66% | **39.99**% |
| | D | 55.04% | 49.96% | 49.55% | 49.79% | 55.85% | 56.53% | **56.73**% | 55.67% |
| | U | 63.25% | 56.53% | 60.93% | 64.12% | 64.32% | 64.84% | 67.58% | **67.69**% |
| | F | 79.88% | 76.79% | 77.67% | 81.40% | 80.69% | 79.54% | **83.25**% | 82.20% |
| AVERAGE | | 59.07% | 55.04% | 55.82% | 57.57% | 59.69% | 59.74% | **61.81**% | 61.39% |
| A,C,G I,O,U | D | 53.60% | 51.52% | 51.86% | 50.94% | 55.54% | 55.69% | 56.71% | **57.07**% |
| | F | 80.80% | 80.58% | 79.70% | 79.40% | 81.62% | 81.66% | 84.30% | **84.50**% |
| AVERAGE | | 67.20% | 66.05% | 65.78% | 65.17% | 68.58% | 68.68% | 70.51% | **70.79**% |

S: source datasets, T: target dataset
A: Aircraft, C: CIFAR100, G: GTSRB, I: ImageNet12, O: Omniglot, U: UCF101, F: Flowers

## 4 RELATED WORKS

Few-shot learning has been studied actively as an effective means for a better understanding of human learning or as a practical learning method only requiring a small number of training examples (Lake et al., 2015; Li et al., 2006). Meta-learning is one of the most popular techniques to solve the few-shot learning problems, which include learning a task-invariant metric space (Snell et al., 2017; Vinyals et al., 2016), learning to optimize (Andrychowicz et al., 2016; Ravi & Larochelle, 2017) or learning good weight initialization (Finn et al., 2017; Nichol et al., 2018) for forthcoming tasks.

Follow-up studies showed that the metric-based meta-learning could be improved further by learning to modulate that metric space in a task-specific manner (Gidaris & Komodakis, 2018; Oreshkin et al., 2018; Qiao et al., 2018; Ye et al., 2018). Similarly, it has been reported that the task-common initial parameters could be refined for a given task producing task-specific initialization (Rusu et al., 2019; Vuorio et al., 2018; Yao et al., 2019).

Recent few-shot learning studies have tried to tackle challenging problems under more realistic assumptions. Some studies dealt with few-shot learning under domain shift between training and testing (Kang & Feng, 2018; Wang & Hebert, 2016). A more realistic evaluation method was proposed for few-shot learning to overcome limitations of the current popular benchmarks including the lack of domain divergence (Triantafillou et al., 2019). One study performed an extensive and fair comparative analysis of well-known few-shot learning methods (Chen et al., 2019).

Our network architecture is inspired by the parameter sharing strategies for multi-task learning (Ruder, 2017) and multi-domain learning with domain-specific adaptation (Rebuffi et al., 2018) because they have been known to lead to efficient parameterization and positive knowledge transfer between heterogeneous entities. Similar to our approach, a few suggestions combined multiple models to benefit from their diversity (Dvornik et al., 2019; Liu et al., 2019; Park et al., 2019). Our work also has something in common with the mixture-of-experts approach (Shazeer et al., 2017) in a sense that a part of a large scale model would be executed conditionally benefiting from a large amount of the learned knowledge at low computational cost.

Our research is also related to domain adaptation or generalization (Ganin et al., 2016; Motiian et al., 2017). However, most of the researches about these topics assume tasks with the same classes in both training and testing whereas our methods do not impose such limitations. Interestingly, some studies showed that the episodic training which is commonly adopted in many few-shot learning techniques, was also useful for domain generalization (Li et al., 2018; 2019).

## 5 CONCLUSION AND FUTURE WORKS

We proposed a new few-shot classification method which is capable of dealing with many different domains including unseen domains. The core idea was to build a pool of embedding models, each of which was diversified by its own modulator while sharing most of parameters with others, and to learn to select the best model for a target task through cross-domain meta-learning. The simplification of the task-specific adaptation as a small classification problem made our selection-based algorithm easy to learn, which in turn helped the learned model to work more effectively for multi-domain few-shot classification. The architecture with one shared model and disparate modulators encouraged our pool to maintain domain-invariant knowledge as well as cross-domain diversity. It helped our algorithms to generalize to heterogeneous domains including unseen ones even when we used one best model solely or all models collectively.

We believe that there is still a large room for improvement in this challenging task. It would be one promising extension to find the optimal way to build the pool without the constraint on the number of models (i.e., one model per dataset) so that it can work even with a single source dataset with large diversity. Soft selection or weighted averaging can be also thought as one of future research directions because a single model or uniform averaging is less likely to be optimal. We can also consider a more scalable extension to allow continual expansion of the pool only by training a modulator for an incoming source domain without re-training all existing models in the pool. Although the number of parameters does not increase much by virtue of the parameter sharing between models, the computational cost in the averaging-based methods needs to be improved over the current linear increase with the number of models.

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

# A DATASETS

In our experiments, we use the *Visual Decathlon Challenge* dataset (Rebuffi et al., 2017) which consists of ten datasets for image classification listed below.

- FGVC-Aircraft Benchmark (Aircraft, A) (Maji et al., 2013)
- CIFAR100 (CIFAR100, C) (Krizhevsky, 2009)
- Daimler Mono Pedestrian Classification Benchmark (DMPCB) (Munder & Gavrila, 2006)
- Describable Texture Dataset (DTD, D) (Cimpoi et al., 2014)
- German Traffic Sign Recognition Benchmark (GTSRB, G) (Stallkamp et al., 2012)
- ImageNet ILSVRC12 (ImageNet12, I) (Russakovsky et al., 2015)
- Omniglot (Omniglot, O) (Lake et al., 2015)
- Street View House Numbers (SVHN) (Netzer et al., 2011)
- UCF101 (UCF101, U) (Soomro et al., 2012)
- Flowers102 (Flowers, F) (Nilsback & Zisserman, 2008)

The categories and the number of images of each dataset as well as the image sizes are significantly different. All images have been resized isotropically to $72 \times 72$ pixels so that each image from various domains has the same size.

Daimler Mono Pedestrian Classification has only 2 classes, pedestrian and non-pedestrian. We excluded it from our experiments as we are considering 5-way classification tasks. SVHN was also excluded since SVHN has only 10 digit classes from 0 to 9, which were too few to split for meta-training and meta-testing. To use the remaining eight datasets for multi-domain few-shot classification, we divide the examples into roughly 70% training, 15% validation, and 15% testing classes. For ILSVRC12, we follow the split of Triantafillou *et al.* (Triantafillou et al., 2019) to adopt class hierarchy, and we use random class splits for other datasets. The number of classes at each split is shown in Table A1. We only use train and validation sets of the Visual Decathlon because the labels of the test set is not publicly available.

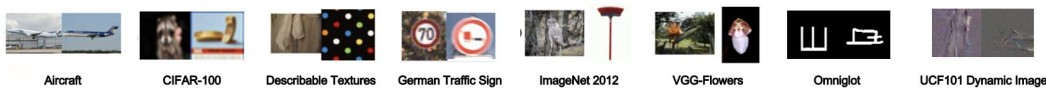

Table A1: The details of datasets used in our experiments.

| DATASET | # DATA | # CLASSES | SPLITS | | |
| --- | --- | --- | --- | --- | --- |
| | | | TRAIN | VAL | TEST |
| AIRCRAFT | 6667 | 100 | 70 | 15 | 15 |
| CIFAR100 | 50000 | 100 | 70 | 15 | 15 |
| DTD | 3760 | 47 | 32 | 7 | 8 |
| GTSRB | 39209 | 43 | 30 | 6 | 7 |
| IMAGENET12 | 1281167 | 1000 | 712 | 158 | 130 |
| OMNIGLOT | 25968 | 1623 | 1136 | 243 | 244 |
| UCF101 | 9537 | 101 | 70 | 15 | 16 |
| FLOWERS | 2040 | 102 | 70 | 16 | 16 |

# B ARCHITECTURES

Figure 2 shows the architecture of the embedding network $f_E(\cdot; \theta, \alpha_i)$, which processes an input image and produces a 512-dimensional embedding vector. The embedding network is based on the

|  | CONVOLUTION 1X1 | CHANNEL-WISE TRANSFORM |
|---|---|---|
| MODULATORS $\{\alpha_i\}_{i=1}^8$ | 9,795,584 | 61,440 |
| BASE NETWORK $\theta$ | 11,176,512 | 11,176,512 |
| SELECTION NETWORK $\phi$ | 66,696 | 66,696 |
| SUM | 21,038,792 | 11,304,648 |

Table A2: The comparison of the number of parameters for convolution $1 \times 1$ and channel-wise transform modulators. This is the case when the number of source domains is 8.

ResNet-18 architecture (He et al., 2016), which consists of one convolutional layer with 64 $7 \times 7$ filters followed by 4 macro blocks, each having 64-128-256-512 $3 \times 3$ filters. Figure 2(a) and Figure 2(b) depict how the base network is modulated by the convolution $1 \times 1$ modulator and the channel-wise transform modulator, respectively. These modulators are placed within each residual block of the macro blocks, same as the previous works in (Rebuffi et al., 2018) and (Perez et al., 2018).

The number of parameters for two modulators are shown in Table A2. The values on the first row are the number of modulator parameters that are additionally applied to the embedding network. Note that the channel-wise transform modulator has much fewer parameters than the convolution $1 \times 1$ modulator. In particular, the channel-wise transform modulator has negligible number of parameters compared to that of the base network, i.e., ResNet-18. For each embedding model, the convolution $1 \times 1$ modulator has about 10% of the number of parameters that the base network has whereas the channel-wise transform modulator requires only less than 1%.

The selection network $f_S(\cdot; \phi)$ is a two-layered MLP (multi-layer perceptron) network, which receives an embedding vector produced by the embedding network as an input and performs the best model index prediction. Two layers are a linear layer of $512 \times 128$ and a linear layer of $128 \times (M+1)$, where $M$ is the number of source domains.

## C  TRAINING DETAILS

Algorithm 1 describes the overall training procedure to construct the model pool and the selection network. Although we trained three components in a sequential manner, joint training of these components seems to make sense also.

For the fair comparison with *Fine-tune* method, we also apply algorithm-specific refinement at meta-testing time, inspired by 'further adaptation' in (Chen et al., 2019), to all other algorithms including ours. A linear classifier is placed on top of the embedding network of the *ProtoNet*, the self-attention module of the *FEAT* or the modulated embedding network of our models. During meta-testing, other parameters are fixed and the classifier is fine-tuned using the support examples for 100 iterations per episode. In case of *FEAT*, the classifier is trained for 100 epochs per query example not per episode because *FEAT* modulates a representation space for each query. We also adjust the number of adaptation of the *ProtoMAML* to 100 for the better task-adaptation as done in (Chen et al., 2019).

The hyperparameters including the learning rate are selected by grid search based on the validation accuracy. For *FEAT* and *ProtoMAML*, Adam optimizer is used for training and the learning rate and weight decay are set to be 0.0001. Other models are also trained using Adam optimizer with the learning rate 0.001 but without any regularization including the weight decay.

In *Simple-Avg*, each embedding model is trained on a separate source dataset following the method proposed in the Prototypical Networks (Snell et al., 2017). We can obtain a little higher performance with this approach than training each model following the standard supervised learning procedure.

All reported test accuracy values are averaged over 600 test episodes with 10 queries per class.

---

**Algorithm 1** The overall training procedure

---

**Input:** Training data from $D_S = \{D_{S_i}\}_{i=1}^{M}$, embedding networks $f_E(\cdot)$, a selection network $f_S(\cdot)$.
**Output:** Learned parameters $\theta$, $\{\alpha_i\}_{i=1}^{M}$, $\phi$.
**Step 1: Build a base network**
 1: Build one large classification dataset $(x_{agg}, y_{agg})$ by aggregating all classes from $D_S$.
 2: Learn $\theta$ by optimizing $f_E(x; \theta, \alpha_0)$ for the aggregated dataset ($\alpha_0$: no modulation).
**Step 2: Add modulators through intra-domain episodic training**
 1: **while** not converged **do**
 2:    Sample one domain $D_{S_i}$ from $D_S$, then sample one episode $(S, Q)$ from $D_{S_i}$.
 3:    Learn $\alpha_i$ by optimizing $f_E(x; \theta, \alpha_i)$ for $(S, Q)$ while keeping $\theta$ fixed.
 4: **end while**
**Step 3: Build a selection network through cross-domain episodic training**
 1: **while** not converged **do**
 2:    Sample one domain $D_{S_i}$ from $D_S$, then sample one episode $(S, Q)$ from $D_{S_i}$.
 3:    Get a task representation $z_{task}$ by averaging embedding vectors of $S$ from the base network.
 4:    Measure accuracies of $M + 1$ available embedding models $\{f_E(x; \theta, \alpha_i)\}_{i=0}^{M}$ for $(S, Q)$.
 5:    Set the best model index $y_{sel}$ to the index of the model with the highest accuracy.
 6:    Learn $\phi$ by training $f_S(z_{task}; \phi)$ so as to predict $y_{sel}$ for $(S, Q)$.
 7: **end while**

---

# D  ADDITIONAL EXPERIMENTAL RESULTS

## D.1  RESULTS WITHOUT FURTHER ADAPTATION

We present the experimental results when we do not apply the further adaptation scheme introduced in (Chen et al., 2019). Specifically, *ProtoNet*, *FEAT*, and our models are tested without additional linear classifiers $f_c(\cdot; \psi)$. The number of parameter update steps in *ProtoMAML* is reduced to 3, which is not enough to have the models fine-tuned. Tables A3 and A4 show the results tested on seen and unseen domains, respectively. We can see that accuracy drops in almost all cases compared to corresponding cases with further adaptation whose results are in Section 3 in the main text, but our models generally do better than other methods in any experimental settings.

Table A3: 5-way 5-shot classification accuracy on seen domains without further adaptation.

| | METHODS | | | | | | | | |
| TARGET | F-T | PROTO | FEAT | PMAML | S-AVG | DOS | DOS-CH | DOA | DOA-CH |
|---|---|---|---|---|---|---|---|---|---|
| A | 39.53% | 49.61% | 58.53% | **66.39%** | 37.04% | 62.86% | 59.39% | 42.82% | 42.91% |
| C | 53.94% | 57.48% | 53.85% | 59.37% | 48.60% | 69.94% | **71.57%** | 62.96% | 64.37% |
| D | 55.68% | 53.51% | 52.91% | 50.81% | 42.98% | **58.65%** | 56.50% | 58.43% | 56.54% |
| G | 78.67% | 86.21% | 92.74% | 95.22% | 74.39% | 96.49% | **96.63%** | 82.93% | 83.31% |
| I | 54.64% | 52.23% | 49.91% | 46.80% | 41.69% | 55.73% | 55.49% | **57.83%** | 57.42% |
| O | 92.96% | 96.20% | 93.18% | 98.98% | 94.44% | 98.86% | **98.80%** | 95.54% | 96.70% |
| U | 62.78% | 66.31% | 64.76% | 66.17% | 59.15% | **69.78%** | 69.45% | 69.68% | 68.93% |
| F | 79.92% | 83.46% | 74.80% | 75.84% | 69.98% | 83.60% | **84.12%** | 83.63% | 83.28% |
| AVERAGE | 64.76% | 68.13% | 67.58% | 69.95% | 58.53% | **74.49%** | 73.99% | 69.23% | 69.18% |

A: Aircraft, C: CIFAR100, G: GTSRB, I: ImageNet12, O:Omniglot, U:UCF101, F:VGG-Flowers
F-T: Fine-tune, Proto: ProtoNet, PMAML: ProtoMAML, S-Avg: Simple-Avg

## D.2  TEST ACCURACY IN A NUMERIC FORM

In Table A5 and A6, we present the exact values of test accuracy with the largest number of source domains, which are shown in Figures 4 and 5 in the main text.

## D.3  COMPARATIVE ANALYSIS ABOUT THE AVERAGING METHODS

As an effort for better understanding the averaging methods, we investigate how each model in the pool contributes to the final prediction. Figures A1(a) and A1(b) show how many correct predictions

Table A4: 5-way 5-shot classification accuracy on unseen domains without further adaptation.

| | METHODS | | | | | | | | |
|---|---|---|---|---|---|---|---|---|---|
| TARGET | F-T | PROTO | FEAT | PMAML | S-AVG | DOS | DOS-CH | DOA | DOA-CH |
| A | 36.40% | 35.42% | 33.30% | 32.87% | 33.48% | 34.73% | 35.63% | 36.93% | **37.28**% |
| C | 53.26% | 55.17% | 52.60% | 58.01% | 49.23% | 55.79% | 50.09% | 58.30% | **59.22**% |
| D | **55.18**% | 51.03% | 50.37% | 45.46% | 45.99% | 54.11% | 48.81% | 54.88% | 54.37% |
| G | 78.01% | 76.33% | 75.79% | **78.81**% | 69.49% | 77.03% | 77.18% | 77.36% | 77.31% |
| I | 34.81% | 32.90% | 33.50% | 36.46% | **37.79**% | 34.34% | 32.73% | 34.36% | 35.27% |
| O | 92.69% | 92.64% | 91.99% | 83.80% | 91.45% | 93.68% | 91.99% | **94.95**% | 94.84% |
| U | 62.16% | 59.74% | 58.54% | 58.40% | 59.06% | 62.04% | 62.21% | 65.93% | **67.25**% |
| F | 81.00% | 79.42% | 80.82% | 69.47% | 75.39% | 81.85% | 82.69% | 82.71% | **83.89**% |
| AVERAGE | 61.69% | 60.33% | 59.62% | 57.91% | 57.74% | 61.70% | 60.17% | 63.19% | **63.68**% |

A: Aircraft, C: CIFAR100, G: GTSRB, I: ImageNet12, O:Omniglot, U:UCF101, F:VGG-Flowers
F-T: Fine-tune, Proto: ProtoNet, PMAML: ProtoMAML, S-Avg: Simple-Avg

Table A5: Few-shot classification accuracy on seen domains. All eight domains were used for training.

(a) 5-way 1-shot test accuracy.

| | METHODS | | | | | | | | |
|---|---|---|---|---|---|---|---|---|---|
| TARGET | F-T | PROTO | FEAT | PMAML | S-AVG | DOS | DOS-CH | DOA | DOA-CH |
| A | 27.90% | 39.05% | 45.39% | **53.03**% | 30.45% | 49.14% | 44.02% | 32.93% | 32.63% |
| C | 36.08% | 38.65% | 34.51% | 40.41% | 33.22% | **50.41**% | 49.58% | 42.84% | 43.90% |
| D | 38.07% | 38.77% | 34.16% | 38.45% | 29.52% | **41.15**% | 40.53% | 39.35% | 38.86% |
| G | 54.37% | 78.57% | 84.85% | 88.85% | 67.97% | **91.08**% | 89.02% | 71.60% | 70.68% |
| I | 37.92% | 38.17% | 33.44% | 33.53% | 31.03% | 40.60% | 40.76% | 41.55% | **41.56**% |
| O | 75.52% | 86.35% | 81.44% | 94.30% | 86.88% | **95.37**% | 95.25% | 87.69% | 89.74% |
| U | 44.46% | 48.40% | 46.63% | 51.46% | 40.36% | 50.16% | **53.19**% | 50.03% | 49.22% |
| F | 59.32% | 62.11% | 54.05% | 57.20% | 49.62% | 65.36% | **66.94**% | 65.34% | 65.93% |
| AVERAGE | 46.71% | 53.76% | 51.81% | 57.15% | 46.13% | **60.41**% | 59.91% | 53.92% | 54.06% |

A: Aircraft, C: CIFAR100, G: GTSRB, I: ImageNet12, O:Omniglot, U:UCF101, F:VGG-Flowers
F-T: Fine-tune, Proto: ProtoNet, PMAML: ProtoMAML, S-Avg: Simple-Avg

(b) 5-way 5-shot test accuracy.

| | METHODS | | | | | | | | |
|---|---|---|---|---|---|---|---|---|---|
| TARGET | F-T | PROTO | FEAT | PMAML | S-AVG | DOS | DOS-CH | DOA | DOA-CH |
| A | 39.53% | 49.82% | 57.62% | 65.89% | 40.79% | **62.65**% | 59.31% | 47.09% | 46.51% |
| C | 53.94% | 57.45% | 53.30% | 60.90% | 49.86% | 70.73% | **71.29**% | 64.21% | 65.70% |
| D | 55.68% | 52.43% | 52.49% | 50.94% | 44.42% | 59.27% | 56.80% | **59.68**% | 57.74% |
| G | 78.67% | 89.28% | 92.19% | 95.25% | 81.59% | 96.82% | **96.91**% | 89.38% | 89.24% |
| I | 54.64% | 51.86% | 48.79% | 47.26% | 42.35% | 54.95% | 55.36% | **58.04**% | 56.74% |
| O | 92.96% | 96.65% | 93.10% | **98.92**% | 95.36% | 98.78% | 98.88% | 96.53% | 97.55% |
| U | 62.76% | 66.02% | 64.77% | 66.09% | 62.66% | 70.32% | 70.46% | 69.93% | **70.57**% |
| F | 79.92% | 83.30% | 74.38% | 75.17% | 72.00% | 84.90% | 84.68% | **85.11**% | 84.46% |
| AVERAGE | 64.76% | 68.35% | 67.08% | 70.05% | 61.13% | **74.80**% | 74.21% | 71.25% | 71.06% |

A: Aircraft, C: CIFAR100, G: GTSRB, I: ImageNet12, O:Omniglot, U:UCF101, F:VGG-Flowers
F-T: Fine-tune, Proto: ProtoNet, PMAML: ProtoMAML, S-Avg: Simple-Avg

are made by each model with the *Simple-Avg* and our *DoA* methods respectively given 50 queries per episode for 40 episodes.

The measured numbers show that the individual models of our *DoA* perform better than those in the *Simple-Avg*, which explains the higher performance of the proposed method partly. Additionally, we can observe that major contributors (i.e., the models with higher accuracy) tend to change every episode in our *DoA* whereas only two models seem to play dominant roles regardless of the given episode. This implies that our method for constructing the model pool provides the averaging model with more beneficial diversity.

Table A6: Few-shot classification accuracy on unseen domains. Seven domains other than the target domain were used for training.

(a) 5-way 1-shot test accuracy.

| | METHODS | | | | | | | | |
|---|---|---|---|---|---|---|---|---|---|
| TARGET | F-T | PROTO | FEAT | PMAML | S-AVG | DOS | DOS-CH | DOA | DOA-CH |
| A | 26.78% | 26.88% | 27.50% | 25.38% | 26.05% | 27.05% | 26.76% | 27.82% | **28.01%** |
| C | 36.43% | 35.35% | 37.63% | 30.04% | 31.57% | 34.56% | 30.93% | **40.05%** | 39.78% |
| D | 37.73% | 34.25% | 35.18% | 30.61% | 29.06% | 37.26% | 36.56% | **38.97%** | 37.24% |
| G | 53.40% | 54.07% | 47.83% | 50.68% | 45.10% | 55.77% | 56.96% | **58.81%** | 57.28% |
| I | 27.18% | 26.03% | 25.97% | 27.18% | 27.53% | 25.58% | 26.45% | 27.56% | **27.88%** |
| O | 75.11% | 68.58% | 75.09% | 55.28% | 71.16% | 70.73% | 71.26% | **80.26%** | 80.06% |
| U | 42.11% | 39.06% | 38.12% | 37.16% | 34.63% | 39.62% | 43.04% | **46.20%** | 45.88% |
| F | 59.32% | 55.94% | 56.34% | 45.93% | 46.02% | 56.38% | 58.98% | **64.56%** | 63.51% |
| AVERAGE | 44.76% | 42.52% | 42.96% | 37.78% | 38.89% | 43.37% | 43.87% | **48.03%** | 47.45% |

A: Aircraft, C: CIFAR100, G: GTSRB, I: ImageNet12, O:Omniglot, U:UCF101, F:VGG-Flowers
F-T: Fine-tune, Proto: ProtoNet, PMAML: ProtoMAML, S-Avg: Simple-Avg

(b) 5-way 5-shot test accuracy.

| | METHODS | | | | | | | | |
|---|---|---|---|---|---|---|---|---|---|
| TARGET | F-T | PROTO | FEAT | PMAML | S-AVG | DOS | DOS-CH | DOA | DOA-CH |
| A | 36.40% | 36.12% | 34.81% | 33.32% | 34.75% | 36.68% | 37.87% | 39.03% | **39.41%** |
| C | 53.26% | 54.98% | 52.40% | 56.14% | 48.52% | 57.25% | 51.11% | 59.83% | **60.99%** |
| D | 55.18% | 51.91% | 51.90% | 45.40% | 45.12% | 55.49% | 51.65% | **56.47%** | 55.25% |
| G | 78.01% | 80.23% | 76.20% | 79.96% | 76.29% | 80.98% | 81.21% | 81.90% | **82.00%** |
| I | 34.81% | 32.42% | 31.27% | 36.50% | 38.15% | 34.16% | 32.68% | 35.33% | **36.31%** |
| O | 92.69% | 92.87% | 89.93% | 83.29% | 91.79% | 94.08% | 92.34% | 95.29% | **95.71%** |
| U | 62.16% | 60.32% | 57.53% | 57.49% | 59.12% | 62.33% | 62.31% | 67.57% | **67.92%** |
| F | 81.00% | 79.81% | 76.62% | 69.55% | 76.57% | 83.39% | 82.37% | 84.41% | **84.58%** |
| AVERAGE | 61.69% | 61.08% | 58.83% | 57.71% | 58.79% | 63.05% | 61.44% | 64.98% | **65.27%** |

A: Aircraft, C: CIFAR100, G: GTSRB, I: ImageNet12, O:Omniglot, U:UCF101, F:VGG-Flowers
F-T: Fine-tune, Proto: ProtoNet, PMAML: ProtoMAML, S-Avg: Simple-Avg

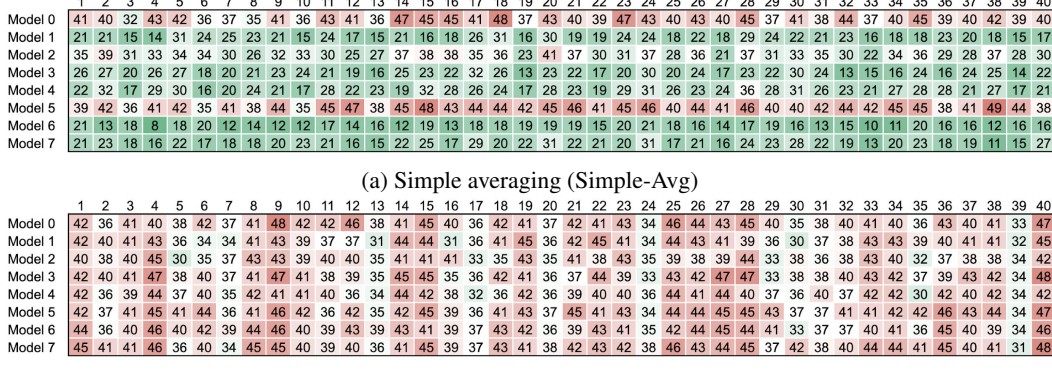

(a) Simple averaging (Simple-Avg)

(b) Proposed averaging (DoA)

Figure A1: Contributions of individual models in model averaging methods.

# E   THE LOSS FUNCTION FOR THE SELECTION NETWORK

Equation (3) shows the loss ($loss_{sel}$) used to train the selection network $f_S(\cdot; \phi)$. Here, $acc_m$ is the classification accuracy of the model with the modulator parameterized by $\alpha_m$ in the pool. The accuracy is measured for query examples in a given episode by making a prediction in the same way

with the Protypical Networks (Snell et al., 2017), where the class whose prototype is the closest to the embedding vector of a given query example is picked as the final prediction.

$$
\begin{aligned}
z_{task} &= \frac{1}{NK} \sum_{i=1}^{NK} f_E(x_i^s; \theta) \\
\hat{y}_{sel} &= \mathrm{softmax}(f_S(z_{task}; \phi)) \\
y_{sel} &= \underset{m}{\mathrm{argmax}}(\{acc_m(\{x_i^s, y_i^s\}_{i=1}^{NK}, \{x_j^q, y_j^q\}_{j=1}^{T})\}_{m=0}^{M}) \\
loss_{sel} &= \mathrm{cross\_entropy}(\hat{y}_{sel}, y_{sel})
\end{aligned}
\tag{3}
$$

