# OpenReview forum: "Domain-Agnostic Few-Shot Classification by Learning Disparate Modulators"
_ICLR.cc/2020/Conference — Reject_

### Official Review · AnonReviewer1 · 2019-10-11
**Official Blind Review #1**

**Rating:** 3

**Review:**

###Summary###
This paper aims to tackle few-shot classification with many different domains. The idea is to build a pool of embedding models, which are based on the same base network. The models are diversed by their own modulators. The high-level intuition is to let the model pool capture good domain-invariant features by the shared parameters and domain-specific features by the selection network, which is desirable to represent the complex cross-domain task distribution, without a significant increase in the number of parameters.

The model includes the embedding networks f_E and a selection network f_S. The overall training process of the proposed method is:
(1) Train the base embedding network f_E with the aggregated dataset from multiple domains.
(2)Sample one domain to optimize the corresponding embedding model.
(3) Build a selection network through cross-domain episodic training. The task representation is calculated by average the embedding vectors from the base network, which resembles the prototype. Then the model with the highest accuracy on the query set is selected to compute the labels for the target domain.

For the embedding model f_E, the paper provides two architectures, one with convolution 1x1 and the other with Channel-wise transform, denoted as DoS and DoS-CH, respectively. Instead of selecting the model which has the highest accuracy, the averaging model generates an output by averaging class prediction probabilities of all constituent models, denoted by DoA and DoA-CH, respectively.

 The baseline used in this paper includes Fine-tune, Simple-Avg, ProtoNet, FEAT, ProtoMAML.

The paper performs extensive experiments on a batch of datasets, including Aircraft, CIFAR100, DTD, GTSRB, ImageNet12, Omniglot, SVHN, UCF101, and Flowers. The 5-way 1 shot and 5 way 5 shot experimental results demonstrate that the proposed DoS and DoS-CH can outperform other baselines in the "see domains" setting. However, the results on "unseen domains" experiments are worse than the averaging baselines (DoA, DoA-Ch).

The paper also surveys the few-shot classification results on a varying number of source domains to show that DoA and DoA-ch are robust to deal with different settings.

### Novelty ###

The paper composes of two sub-network, with one baseline network to extract the commonsense knowledge for different datasets and another selection network to select the best model from the model pools for each specific domain.

The idea of leveraging multiple modulators for domain-agnostic image recognition is interesting and heuristic, thus the proposed framework shows some novelty.

###Clarity###

Overall, the paper is readable and logically clear. The images are well-presented and well-explained by the captions and the text.

While this paper misses many prior works both in the track of domain-agnostic learning and few-shot learning. I would recommend the authors to the following materials:
a). Domain Agnostic Learning with Disentangled Representations, ICML 2019. https://arxiv.org/pdf/1904.12347.pdf
b). Generalizing from a Few Examples: A Survey on Few-Shot Learning,
https://arxiv.org/pdf/1904.05046.pdf

There also exist some grammar mistakes and typoes in the paper. It will be better to revise and polish the paper.

###Pros###

1) The paper proposes a framework that includes two parts, i.e. the base network and the selection network. The idea is to make deep models more robust to different image domains, which is interesting and heuristic.
2) The paper provides extensive experiments on a wide range of datasets. The experiments demonstrate the effectiveness of the proposed method.
3) The paper is applicable to many practical scenarios since the data from the real-world application is complicated.

###Cons###

1) The critical component of the proposed method is the selection network. However, the experiments on the "novel domains" show that the proposed DoS and DoS-Ch works worse than just averaging the outputs of the models in the model pool.
2) The paper should explain the details about the baseline experiments. Since the proposed models have much more parameters than the baselines, what's the effect of the auxiliary parameters? Is the comparison between the baselines and the proposed method fair?
3) The presentation of the proposed paper should be polished. Many critical techniques used in this paper are not well-explained, such as the ProtoNet. There exist some grammar mistakes and typos that need a revision.


Based on the summary, cons, and pros, the current rating I am giving now is weak reject. I would like to discuss the final rating with other reviewers, ACs.



**Experience Assessment:**

I have published one or two papers in this area.

**Review Assessment: Checking Correctness Of Derivations And Theory:**

N/A

**Review Assessment: Checking Correctness Of Experiments:**

I carefully checked the experiments.

**Review Assessment: Thoroughness In Paper Reading:**

I read the paper thoroughly.

---

> ### Author Response · Authors · 2019-11-12
> **Response to the review #1**
>
> Thank you for your valuable feedback.
>
> As the reviewer pointed out, our selection network is not as effective for novel domains as the averaging methods. Since the current selection network chooses only a single best model from the pool, the chosen model might be suboptimal to represent a novel domain which is not close to any source domains. In this case, we hypothesize that an ensemble method could lead to better generalization to unseen domains because the ensemble learning reduces model variance in general. Our major contribution is to introduce a simple and effective way to build a model as a component of the ensemble, i.e., hard parameter sharing with domain-specific modulator, to deal with novel domains. Averaging those models showed much higher performance with much smaller number of parameters than the “Simple-Avg”, averaging models each of which was trained independently on a particular source domain without sharing. These results signify the effectiveness of the proposed method in forming the ensemble.
>
> As for the second concern, we detail the number of parameters of all methods under comparison. We used the ResNet-18 as the base network for all the methods including ours. No additional parameters are required for “Fine-tune”, “ProtoNet” and “ProtoMAML” while “FEAT” and our methods need more parameters for an attention module and  a modulator module in addition to the base network respectively. Their numbers of parameters and relative sizes compared to the base network are as follows.
> - Fine-tune, ProtoNet, ProtoMAML: 11,176,512 (100.0%)
> - DoS-Ch, DoA-Ch (w/ 8 source domains): 11,304,648 (101.1%)
> - FEAT: 12,228,160 (109.4%)
> - DoS, DoA (w/ 8 source domains): 21,038,792 (188.2%)
> - Simple-Avg (w/ 8 source domains): 89,412,096 (800.0%)
> Our methods using channel-wise transform modulators (“DoS-Ch”, “DoA-Ch”) have only 1.1% more parameters than the methods using the base network. Our methods using 1x1 convolution modulators (“DoS”, “DoA”) require more parameters, however, which are still much fewer than that of the ensemble method with the same number of independently trained models (“Simple-Avg”).
>
> Lastly, we will add more detailed explanation of the ProtoNet-like training and correct grammatical mistakes and typos in our revision. We will also add the recommended previous works, “Domain Agnostic Learning with Disentangled Representations” and “Generalizing from a Few Examples: A Survey on Few-Shot Learning”, in our revision.

---

### Official Review · AnonReviewer3 · 2019-10-23
**Official Blind Review #3**

**Rating:** 3

**Review:**

In this paper, the authors proposed to extend meta learning for few-shot classification to the multi-domain setting. The problem looks interesting, but the proposed method is incremental.

Specifically, there are three main components of the proposed method in training: 1) a base network, 2) a model pool, and 3) a selection network. The base network is trained by simply combing training data from all the domains, which is the most common way to learn a shared model across different domains. The model pool is constructed highly based on existing works [Snell et al., 2017; Rebuffi et al., 2018], and thus is out of novelty. The main technical contribution should lie on the selection network, where a key research issue is how to represent different tasks. However, in this work, the authors just simply used the mean of outputs of the base network over all instances of a specific task to represent the task. Why is this strategy good for task representation? The authors failed to explain it. Neither theoretical nor empirical studies are provided. Therefore, the motivation behind this is unclear and not convincing.

In testing or inference, the authors proposed to use a class-prototype based approach or a weighted average of ensemble outputs to make predictions on the target-domain instances. Theses techniques are indeed widely used in the literature, which are again not novel.

In summary, though the problem studied in this paper is interesting, the proposed method is incremental, and has limited novelty contributions.

**Experience Assessment:**

I have published one or two papers in this area.

**Review Assessment: Checking Correctness Of Derivations And Theory:**

N/A

**Review Assessment: Checking Correctness Of Experiments:**

I assessed the sensibility of the experiments.

**Review Assessment: Thoroughness In Paper Reading:**

I read the paper at least twice and used my best judgement in assessing the paper.

---

> ### Author Response · Authors · 2019-11-12
> **Response to the review #3**
>
> Thank you for your valuable feedback.
>
> We believe that the novelty of our method lies in the way of combining existing techniques to solve few-shot classification tasks in the multi-domain setting. Although each technique has been proposed independently with a different purpose, they have been never used together to solve the raised problem to the best of our knowledge. In addition, we showed its effectiveness empirically through various experiments.
>
> Regarding the task representation, we hypothesize that the task representation is an output from a permutation invariant function of all support set examples. As “Deep Sets” (M. Zaheer et al., NIPS 2017) proved, the permutation invariant function can be modeled as nonlinear transform of a sum of all element-wise representations. Actually, the first layer of our selection network and the mean calculation in our method correspond to the nonlinear transform and the sum operation mentioned in Deep Sets. It implies that our architecture is sufficient to learn good permutation invariant representation of the task. Although the element-wise representation can be learned independently of the base network, we decided to report only results with the base network because having a separate network showed only a marginal impact in the accuracy while increasing the number of parameters significantly.

---

### Official Review · AnonReviewer2 · 2019-10-23
**Official Blind Review #2**

**Rating:** 3

**Review:**

In this paper, the authors proposed to address the few-shot learning problem, especially for the cross-domain setting where a newly coming task originates from a different distribution (or in this work implemented by sampling from an unseen dataset). Basically, the authors constructed a model zoo based on source datasets at hand, and learned an “argmax” meta-selector which takes embedding of a task as input and outputs the model selection index. The idea is very intuitive, and the implementation is kind of an incremental combination of (Rebuffi et al., 2018) building models for multi-tasks and (Oreshkin et al., 2018) tailoring based on task embeddings.

Pros:
-	The problem that this work aims to address, i.e., domain heterogeneity, is significant.
-	The author investigated several variants of the proposed method, varying the architecture of the modulator and the inference scheme.
-	The paper makes clear points and is quite easy to follow.

Cons:
-	The concern at first priority is the practicability of the proposed method. The authors likely misunderstand why almost all existing SOTA algorithms modulate parameters/activations themselves – it is because of their advantages of easily being efficiently updated across tasks. However, the proposed method has to store an increasing number of models as a task comes, which takes huge storage cost. Also, when a new task arrives, will all models be trained from scratch to enforce the feature extractor to also be shared by this new task? It is an economically infeasible solution for meta-learning/few-shot learning, to my best knowledge.
-	Actually while I was reading the paper from the beginning, I was expecting a meta-scheme that learns the weights of the models in the pool. In that case, the inference scheme of averaging in Eqn. (2) could be more intuitively correct, by paying more attention to those models which are similar.
-	The third concern comes from the empirical results.
    o	Baselines: the two baselines compared, FEAT and ProtoMAML, are actually un-published. The results of a handful of SOTA algorithms recently published, including (Oreshkin et al., 2018) and (Yao et al., 2019), should definitely be incorporated. I do not see any specific part in your setup, and I cannot understand why (Oreshkin et al., 2018) does not converge. According to my experiences, it is quite easy to converge. Besides, (Yao et al., 2019) is specifically designed to tackle the cross-domain (for an unseen dataset) setting.
    o	How can you validate the effectiveness of the task embedding? It is better to conduct ablation studies to consider those like autoencoder embeddings.
    o	Could you introduce the setting of fine-tuning in more details? I am confused why fine-tuning seems always superior than ProtoNet?
    o	Could you give more discussion on the results in Table 1? As the number of source datasets increases, an effective meta-model of course should contribute more to the target dataset, while it is not in this work. Does this signify that the proposed method is not that effective?


**Experience Assessment:**

I have published one or two papers in this area.

**Review Assessment: Checking Correctness Of Derivations And Theory:**

I carefully checked the derivations and theory.

**Review Assessment: Checking Correctness Of Experiments:**

I carefully checked the experiments.

**Review Assessment: Thoroughness In Paper Reading:**

I read the paper thoroughly.

---

> ### Author Response · Authors · 2019-11-12
> **Response to the review #2**
>
> Thank you for your valuable feedback.
>
> Regarding the first concern about the practicability, we would like to stress that our solution does not require building one embedding model for each coming task. The model pool construction is being performed only during meta-training. When we encounter a new few-shot task at meta-testing time, we use the trained models as is to solve that task without training any additional model. Even when we are performing episodic training with various random tasks at meta-training time, the number of models does not increase because we assume building one model per source domain not per task. Since the number of source domains, each of which corresponds to one dataset in our setup, is predefined, the size of the pool never changes throughout both meta-training and meta-testing. As shown in Table A2, the whole pool which has 8 constituent models is only 1.1% and 88.2% larger than a single model (i.e., the base network) with the channel-wise transform modulators and 1x1 convolution modulators, respectively.
>
> For now, our methods either choose only one model from the pool or average outputs of all models for inference. We agree with the reviewer in that better solutions can be found by learning to assign task-adaptive weight to each model beyond the current two extreme selection methods.
>
> As for the baseline, we chose ones with the highest classification accuracies in a typical setup (“FEAT”) and a multi-domain setup (“ProtoMAML”) rather than sticking to the officially published ones. Although “TADAM” (Oreshkin et al.) showed promising results in the typical few-shot learning setup, we failed to achieve the accuracy comparable to other methods in the multi-domain setup. It might be because it was too hard to capture a wide range of task distribution across domains with a single parametric function that TADAM proposed. Since “Hierarchically Structured Meta-learning (HSML)” (Yao et al., 2019) deals with multi-domain setup, we also agree that “HSML” can be a good baseline, so we will try to add comparison results in our revision.
>
> Regarding the task embedding, we hypothesize that the task embedding is an output from a permutation invariant function of all support set examples. As “Deep Sets” (Zaheer et al., NIPS 2017) proved, the permutation invariant function can be modeled as nonlinear transform of a sum of all element-wise representations. Actually, the first layer of our selection network and the mean calculation in our method correspond to the nonlinear transform and the sum operation mentioned in Deep Sets. It implies that our architecture is sufficient to learn good permutation invariant representation of the task.
>
> Both “Fine-tune” and “ProtoNet” are pre-trained with one virtual task which classifies all classes in all available source datasets by performing typical supervised learning on one virtual dataset formed by aggregating all source datasets. In “Fine-tune”, we retrain only the final linear layer with the support set of a given task while freezing the rest of the network. However, in “ProtoNet”, we update the whole network starting from the pre-trained network by performing episodic training following the common meta-learning procedure. In our results, “ProtoNet” tends to outperform “Fine-tune” in seen target domains while “Fine-tune” seems slightly better on unseen target domains. Some previous studies have already revealed that “Fine-tune” was a strong baseline despite its simplicity ( “A Closer Look at Few-shot Classification” (W. Chen et al., ICLR 2019), “Meta-Dataset: A Dataset of Datasets for Learning to Learn from Few Examples
> (E. Triantafillou et al., arXiv:1903.03096)). It does not seem surprising that “Fine-tune” often performs better than others in some cases.
>
> As the reviewer pointed out, the results in Table 1 show that more source domains do not guarantee higher performance when we consider a particular target domain. Similar trends are also shown in other baseline methods. It shows that there is a large room for improvement in avoiding negative transfer in the multi-domain environment. However, the model trained with more source domains is still useful because it can process more domains as “seen” domains without a significant increase of the model size. For example, if our model has been trained on ‘C’ & ‘I’ with ‘DoS-Ch’, this model would show 82.65% test accuracy on ‘G’ (refer to 5th row & 8th column in Table 2). But, if we have another model trained on ‘C’, ‘I’, ‘G’, & ‘O’ with ‘DoS-Ch’, it would show much higher test accuracy, 96.36%, on the same target domain ‘G’ (refer to 7th row & 8th column in Table 1). It is because ‘G’ has been already seen to the latter model and thus is easier to predict than other unseen domains.

---

### Decision · Program_Chairs · 2019-12-19

**Decision:**

Reject

**Comment:**

This paper addresses the problem of few-shot classification across multiple domains. The main algorithmic contribution consists of a selection criteria to choose the best source domain embedding for a given task using a multi-domain modulator.

All reviewers were in agreement that this paper is not ready for publication. Some key concerns were the lack of scalability (though the authors argue that this may not be a concern as all models are only stored during meta-training, still if you want to incorporate many training settings it may become challenging) and low algorithmic novelty. The issue with novelty is that there is inconclusive experimental evidence to justify the selection criteria over simple methods like averaging, especially when considering novel test time domains. The authors argue that since their approach chooses the single best training domain it may not be best suited to generalize to a novel test time domain.

Based on the reviews and discussions the AC does not recommend acceptance. The authors should consider revisions for clarity and to further polish their claims providing any additional experiments to justify where appropriate.